# Good Anesthesia Practice for Fish and Other Aquatics

**DOI:** 10.3390/biology11091355

**Published:** 2022-09-15

**Authors:** Aurora Brønstad

**Affiliations:** Department of Clinical Medicine, University of Bergen, 5021 Bergen, Norway; aurora.bronstad@uib.no; Tel.: +47-55-97-37-82

**Keywords:** anesthesia, good practice, fish, aquatics

## Abstract

**Simple Summary:**

It is vitally important that fish and other aquatic animals are not at risk of pain, suffering, or distress when they are used in procedures. In addition, many procedures involve taking them out of water, which can be very stressful for them as many species cannot breathe out of water. Proper use of anesthesia can reduce the potential suffering for the fish. However, anesthesia must be performed skillfully to achieve the desired effect and to avoid adverse effects. This paper will focus on important factors to support vital functions in anesthetized animals and will include factors to consider before, during, and after anesthesia. I suggest that these are good anesthetic practices for aquatic animals.

**Abstract:**

Fish and other aquatic animals represent a significant number of species with diverse physiology, size, and housing condition needs. Anesthesia may be necessary for several husbandry procedures as well as treatment of diseases, surgery, or experimental procedures. Choice of drugs and detailed procedures for anesthesia must be adapted to the species in question—there is no “one size fits all” solution. However, there are some basic principles that apply for good anesthetic practice of all animals. These principles include the preparations of animals, personnel, facilities and equipment, monitoring animals under anesthesia, as well as post-anesthetic care to be sure that animals are not lost in the recovery phase. Good anesthesia practice also includes the competence and commitment of personnel involved. Based on professional judgement, key factors will be the focus of this text.

## 1. Introduction

### Why Anesthesia in Aquatic Animals?

There are several reasons for using anesthesia in fish and other aquatic animals. Research has shown that fish experience pain [1,2], suffering, and distress and respond to stressful and noxious stimuli just as other phylogenetically “higher” animals, and this applies for adult fish [3,4], fish larvae [5,6], as well as other invertebrate aquatic animals [7,8,9,10,11]. Furthermore, the administration of analgesic substances with known effects in mammals also modulates responses to painful stimuli in aquatic species [5,7,10,12,13]. Nociception is a useful and necessary capability for self-protection and unquestionable for survival for most creatures. Studies have also suggested affective state and learning abilities for both vertebrate [14] and invertebrate aquatic animals [13,15].

Fish have identical rights to be protected from suffering and pain as other vertebrates such as mammals and birds used in research [16]. Regarding the use of anesthesia in animal studies, the European directive 2010/63 [17], that protects vertebrate animals, states:


*Procedures are carried out under general or local anaesthesia, and that analgesia or another appropriate method is used to ensure that pain, suffering and distress are kept to a minimum.*



*Procedures that involve serious injuries that may cause severe pain shall not be carried out without anaesthesia.*
(Article 14) [17]

A similar statement can be found in The Guide for the Care and Use of Laboratory Animals [18]:


*Pain is a stressor and, if not relieved, can lead to unacceptable levels of stress and distress in animals. … For these reasons, the proper use of anesthetics and analgesics in research animals is an ethical and scientific imperative (page 120)*


Anesthetics may interfere with scientific outcomes, and there may be occasions where a researcher would prefer to avoid the impact of anesthetics. Anesthetic substances may compete with test substances and change distribution, metabolism, or elimination [19,20]. However, as the animal welfare regulations are very clear on this point, exceptions must be based on strong ethical judgment and a harm–benefit analysis [21]. In addition, the decision not to use proper anesthesia, causing pain and stress, may interfere with the outcome of the procedure and for most cases, it will be possible to find alternative solutions that minimize the negative impact on the animals, whether this is other drugs or other refined procedures. 

“Evidence” used to decide on which animals are “sentient” and should be regulated and protected by law is based on several observations, such as:□Functioning of the brain and nervous system; □Complexity of life and behavior;□Learning ability;□Indications of pain or distress;□Studies illustrating the biological basis of suffering, fear, and anxiety;□Indications of awareness based on observations and experimental work.

Regulations must define clear borders of application. However, defining which animals to protect or not is problematic. The more we study different species, the more we discover that more animals meet the criteria listed above, i.e., they have the capability of sentience. Research in non-vertebrate animals showing capabilities for pain, suffering, and distress in other phyla of the animal kingdom have an impact on lawmakers. For example, cephalopods were included in the European directive protecting research on animals [17], or decapods, who were added to the Norwegian law [22] on animal research regulation [23].

These explicit statements that animals must be protected against pain, suffering, and distress reflect a moral stand and can be achieved by taking mitigating actions, such as the use of anesthesia and analgesia. It is argued that we owe aquatic animals’ moral consideration just as we owe respect and moral consideration to all other animal beings, regardless of the taxonomic group [24].

In addition to the regulatory and ethical aspects, there are also several practical reasons for applying anesthesia in fish and other aquatics. Procedures involving handling or restraint usually involve moving them out of water—a threatening condition which causes panic and evokes escape behavior. Proper sedation and anesthesia have the potential to limit the adverse effects caused by the hypothalamic–pitiuary–interrenal (HPI) axis that plays a central role in the physiological stress response. Sedation is indicated to reduce stress and the risk of damage for many procedures or husbandry practices such as handling, crowding, pumping, transport, bath treatments in tanks, well boats or pens, brood stock handling, or as premedication to anesthesia, thus facilitating safe handling and improving fish welfare.

Anesthesia and pain-treatment are compulsory for surgical procedures in fish—even for minor procedures such as “simple injections” [17] (annex VIII). A “simple injection” performed by a competent person is defined as a mild procedure and defines the lower threshold for an intervention in research animals [17]. This definition of a lower threshold, however, is based on land-living animals, which can even be trained for procedures without restrain, by use of animal-centric approaches [25]. This is not be the case for many aquatic animals, where some sort of sedation or anesthesia is necessary to perform procedures safely and without stress. The cumulative impact [26] of all necessary procedures will impact the fish’s welfare, even for “a simple injection”. 

In addition, to ensure a stress- and pain-free death when killing fish, for welfare reasons, such as a humane death or a study’s endpoint, premedication with an anesthetic drug may be preferred, and this is also in compliance with the European directive appendix C [17]. 

There are significant differences in drugs and doses for different aquatic species, and there is no “one size fits all” solution. Safe, efficient anesthesia includes more than the correct dose or concentration. Understanding the impact of basic body functions such as stress and pain mechanisms and the consequences for welfare are all important factors for an anesthesiologist to consider and manage. The principles for good anesthesia practice that apply for land-living animals should also be adhered to when anesthetizing fish and other aquatics. This paper introduces principles of “good anesthetic practice” translated into aquatic animals. The focus will be on critical factors and practices that may sound obvious but that are still forgotten or neglected if not assessed when anesthetizing aquatic animals:


*“Anesthesia is not a dose—good anesthesia practice is performed by skilled anesthesiologist” (personal communication. Professor Andreas Haga, specialist in veterinary anesthesia)*


## 2. Good Anesthetic Practice

### 2.1. Preparation for Anesthesia Procedures

Proper preparation before anesthesia is important, as it minimizes anesthetic complications and ensures satisfactory anesthesia and smooth recovery.

Preparation for anesthesia includes the preparation of all necessary key elements including animals, drugs and test substances, equipment, facilities, and personnel. 

### 2.2. Preparing the Animals and Checking the Health Status 

A welfare assessment checklist may be useful, and at minimum you should check:□Acclimatization;□Appetite, feeding, and defecation;□Swimming pattern and position in water;□Locomotor activity and coordination;□Response to human presence (panic or fatigue);□Interactions with the same species. 

### 2.3. Acclimatization 

Be sure that the animals are acclimated to the experimental conditions and always allow them to recover from metabolic and hormonal changes after transport or other stress. As fish are poikilotherm animals, water temperature will have an impact on their metabolism, and it is important that they are acclimated to the temperature conditions in which the anesthesia procedures will take place. Change in temperature or light regime will increase oxygen consumption, and fish might need time to acclimatize to the changes [27]. Sudden changes in water temperature should be avoided, and adaption to new water temperature regimes should not be faster than 2 °C per 48 h. 

Any deviation in behavior or health condition observed trough the acclimation period should be a warning of anesthetic complications. Fish that are not healthy or fit for purpose will have an increased risk of anesthetic complications. Unhealthy animals need extra care and attention. It should be considered if the fish are ready or fit for anesthetic procedures. Proper acclimatization reduces complications and peri-operative stress. In a new location, fish also need time to habituate to a new environment, staff, and routines. So, even when the animals seem to be in good health, it is in accordance with good scientific practice to always give time for the acclimatization of animals before they participate in any procedures [28].

### 2.4. Appetite, Feeding, and Defecation 

Hungry fish that are fed by hand will usually approach people at the tank side to be ready to receive food. Feed remaining in the fish tank is an indication of reduced appetite in the fish and should be taken as an early warning and an observation that should evoke more investigation, extra attention, and corrections before anesthetizing fish. “Floating feces” are refusals from the intestine and a disease indicator. 

### 2.5. Response to Human Presence (Panic or Fatigue)

The behavior and response when personnel approach the fish tank is important. Unfamiliar human behaviors (new procedures or staff) or equipment may cause panic or escape reactions. Fish will normally respond to human presence, and a lack of response may be an indication of fatigue. An inspection window in the holding tank facilitates observation of behavior and position in the water without lifting the top lid and disturbing the fish. It should be noted if the fish are calm and relaxed or stressed or if they show signs of panic. 

### 2.6. Locomotor Activity and Coordination

Fish that show abnormal swimming behavior may be excluded from a study, but with proper care and time to rest, they may recover. An inspection window gives a better lateral view of the fish and a better impression of abnormalities such as loss of equilibrium or spiral swimming. Skin damage, ulcers, or other damage on fins or eyes can also be observed through an inspection window.

### 2.7. Fasting before Anesthesia

Fasting is species/age/size-dependent in fish. Fasting is recommended before anesthesia [29]. The main purpose is to empty the gastrointestinal system, as vomiting and feces will pollute the water, which will reduce the effect of anesthesia and displace oxygen. Fasting 12–48 h before anesthesia is a general recommendation [29] for group-anesthesia of fish, for example, for vaccination in fish farming. However, science-based recommendations on responsible fasting times that consider fish welfare are lacking [30]. Withdrawal of food may be a stressor for fish, causing aggression and conflicting with ethical and legal obligations to animals [30]. It should therefore be reconsidered if fasting is strictly necessary for all kinds of procedures or studies. If the water pollution problem can be avoided or minimized by reducing fish density and practicing more careful handling to reduce the risk of rupture of a filled stomach, fasting may not be necessary. Feeding a smaller amount of food than what is provided for daily maintenance should also be considered as a compromise and refinement of pre-anesthesia procedures. For some fish species, fasting is a normal part of their life cycle. So, they may naturally tolerate starvation well, and brief food deprivation might not severely harm the health of those fishes.

### 2.8. Preparing Drugs 

All drugs and test substances planned to be used for anesthetic procedure must be checked and cleared for use. Anesthesia drugs must be correctly stored and used within the expiration date. Outdated drugs or drugs stored under wrong conditions may jeopardize animal welfare and have a negative impact on the study [18] (page 122). As a general recommendation, only pharmaceutical-grade drugs should be used to prevent toxic or other unwanted adverse effects ([18] page 31). For fish, many anesthesia substances are available as powders (tricaine, MS222, Finquell vet^®,^ MSD Animal Health Norge AS, Thormøhlens g. 55, 5006 Bergen, Norway ) or as stock solutions (Aquis-S^®^ MSD Animal Health Norge AS, Thormøhlens g. 55, 5006 Bergen, Norway, Isoegenol). Only substances manufactured for the purpose of anesthesia should be used. Similar chemical compounds should not be used for anesthesia, as there may be differences in purity, storage capabilities, and how they react with different water conditions. Furthermore, only use freshly made anesthetic solutions, as mixing the anesthetic drug with water will impact the chemical properties, stability, and storage capacity. Avoid anesthesia under extreme weather conditions outdoors, for example, in direct sun [31]. Direct sun heats the water and reduces oxygen, and some drugs are degraded by sunlight. 

Most anesthetics used for other animals are also effective in fish, for example, ketamine [32] or propofol [33]. However, not all anesthetics can be dissolved in water and administered as a bath solution. Substances that must be administered by injection cause a need for restraint, which will cause extra stress in most aquatic animals. As the fish musculature is compact with the skin tightly adhered around the body, the volume injected for intramuscular subcutaneous injection is rather limited. Larger volumes are tolerated by intraperitoneal injection; however, with this method, there is a risk of injecting the substance into the intestine or another internal organ. Intravenous access is also possible in restrained, immobilized animals. 

Procedures may take more time than expected, so it is necessary to ensure sufficient anesthetic drugs for the whole period, plus unexpected additional requirements. Even when procedures are well planned and organized, unexpected events may happen. You should also have drugs for available emergencies, such as substances used for stimulation of respiration, circulation, or specific antidotes (if available), to be sure that vital functions can be supported if necessary. Examples may include, but are not limited to, adrenalin and atropine. 

For fish intended for human consumption, there are special demands of maximum residue limits (MRL) values to avoid toxic effects in consumers [34]. Within Europe, the European Medicine Agency (EMA) gives recommendations for MRLs for different substances and species. Drugs may be allowed to be used based on MRL values or because MRL values are not considered applicable or necessary based on the risk of toxic effects in consumers. It is important to note that regulations on MRL values are based on current knowledge and that regulations may be changed based on updated knowledge. It is therefore always necessary to check updated regulations, and usually the veterinarian prescribing substances used for animals can advise on this matter. 

The withdrawal time is the time needed before residues of drugs are eliminated or below the upper safety level for animalic products (meat, egg, milk). Metabolism of a substance can impact excretion from the body, and as fish are poikilothermic animals, the withdrawal time will depend both on time (days) and temperature (“degree-days/DD”).

Examples:
**Substance****MRL****DD****Assessment Date**TricaineNot-applicable [34]25 [35]26 August 2022Isoeugenol6 mg/kg [34]2 [35]26 August 2022

### 2.9. Preparing Equipment and Personnel 

Monitoring devices are helpful to check that physiological parameters are held within the optimal range during the whole time of the anesthesia procedure. When monitoring devices are used, they must be checked and calibrated, and correct alarm limits must be set. Opercula beat rate of frequency and ECG are examples of parameters to monitor for regularity and magnitude. Irregular or weak signals may be a warning of impaired vital functions.

All personnel involved in anesthesia must be familiar with procedures, equipment, and techniques, and their responsibilities and tasks must be clarified. Personnel must be trained to understand what critical parameters they should record during anesthesia, how to respond if critical values are reached, and how to implement corrective actions to assure safe anesthesia for all animals.

### 2.10. Personnel Safety 

Personnel safety must be assessed as most anesthetic substances will also affect people. Procedure rooms must be adequately ventilated. Handling stock solutions with high concentrations of active substances is only safe using appropriate personal protective equipment. Personnel should protect their skin and mucous membranes (eyes, nose, mouth) and avoid direct contact with anesthetic solutions. Slippery floors may cause risk of falling and in large-scale systems—a risk of falling into the water and drowning. Pumps and other mechanical installations may cause noise and the need for hearing protection—which may make communication a challenge. 

### 2.11. Introduction of Anesthesia

Anesthesia may be introduced either in the home tank (Figure 1a) or in a special “induction” tank (Figure 1b). 

Moving fish from the home tank or net (Figure 1b) will be a stressor often including netting, pumping, or other restraint of fish. This may cause panic reaction, increased cortisol, and risk of mechanical damage to the fish. The autonomous nervous control of peripheral circulation responds to stress and has a major impact on circulation, as blood distribution during rest and work varies significantly [36]. Adding a sedative or anesthetic solution to the home tank (Figure 1a) will minimize these adverse effects. Inducing anesthesia in a home tank is preferred as it reduces stress, but it is only possible if the circulation of fresh water can be stopped or if water can be treated (filtered and reoxygenated) and recirculated. 

Anesthesia in the home tank will cause anesthesia of all fish in that tank—which is not always necessary or desired, as anesthesia, even following best practices, will have an impact on the physiology of the animal and increased risk for damages to the delicate skin and mucous layer.

### 2.12. Support of the Fish under Anesthesia

Fish that are immersed in anesthetic solutions will show initial excitement, erratic swimming, loss of equilibrium, and muscular tone. Fish will then become inactive and often sink to the bottom of the tank and rest on their back (Figure 2).

The induction period before the fish lose consciousness should be limited to only a few minutes. Too long of an induction time causes more stress and may have a negative effect on metabolism and consumption of energy storage. 

The effect of anesthesia can be evaluated based on loss of posture and righting response, decreased or absent response to noxious stimulus, or depressed ventilation or locomotory activity. If pain-tests cause spasms, the fish is either not deeply anesthetized enough for potential painful procedures or the chosen anesthetic does not have adequate analgetic properties, causing only an apparent sleeping condition but no analgetic effect. The use of neuromuscular blockers cause immobilization; however, all nociceptive processes are still active. The ability to respond to pain will be masked when neuromuscular blockers are used. The animal can experience pain but is paralyzed in a helpless condition, unable to respond to pain. Whenever neuromuscular blockers are used, because of scientific needs and justification, the analgetic effectiveness of the anesthesia regime must first be verified [18]. 

Animals under anesthesia should be monitored to be sure that vital functions are upheld during anesthesia. Basic monitoring and close observation of fish should, as a minimum, include the rate and pattern of respiration and signs of pain or discomfort. The respiration frequency will normally be reduced during anesthesia but should still be regular. Irregular, superficial, gasping, or heaving respiration may indicate problems. Heart rate can be monitored by use of ECG and pulse-rate measurements by pulse oximetry, showing the saturation of oxygen in the blood [37]. In addition, heart rate will be reduced under anesthesia due to autonomous regulation, but the pulse should be regular. Too low of a heart rate may indicate a deep level of anesthesia, and a high heart rate may indicate pain or a circulatory problem. 

Respiratory depression is a common effect of anesthesia, and this may be more severe in animals in poor health. In many fish species, the gills also play an important role in electrolyte exchange. Gill afflictions may reduce gas exchange [38], as, for example, in amoebic gill disease where the free gill surface for oxygen exchange is reduced [39]. It is recommended to increase the oxygen level in the water, rinsing the gills so that the animal is properly oxygenated and to support basic metabolic processes. Attention must be paid, however, to avoid over-saturation of water because this is a risk of air-bubble entrapment and air emboli obstructing circulation. 

For long term anesthesia with the fish out of water, for example for surgery, dehydration, drying, and damage of skin or mucous layer must be prevented by humidification of the surface of the animal. The skin and mucous layer of a fish is an important barrier against pathogens and outer environment factors. Water should be monitored to ensure correct temperature, O_2_ saturation, and pH during the whole procedure. These parameters should have the same optimal conditions in the anesthesia tank in which the fish is acclimatized to avoid adaptive metabolic stress on the fish. 

Responsibility for monitoring animals under anesthesia should be assigned to a person dedicated to keeping focus on the welfare and vital functions of the animal, without too many other distracting duties. Monitoring must be combined with the ability and knowledge to take appropriate corrective actions. Communication with other personnel, such as surgeons, must be clear and confirmative to be sure that the right actions will be taken in a timely manner.

Anesthesia complications—possible factors: □Fish are not acclimatized or healthy and not fit for anesthesia; □Mechanical trauma of animals during handling;□Lack of humidification of the skin when fish are out of water;□Overdose caused by poor rinsing of gills during or after anesthesia;□Anesthetic procedure performed in the presence of sunlight;□Technical issues such as failure in fresh water and oxygen supply. 

### 2.13. After Anesthesia (Post-Anesthetic Care)

Many anesthetic deaths in animals occur because post-anesthetic care is neglected. An appropriate facility with fresh, clean oxygenated water must be prepared for the fish to recover from anesthesia. As for anesthesia induction, it is preferred that the recovery from anesthesia does not take too long. Recovery within 5–10 min is recommended. Recovery water should be of the same quality as the animal has already been acclimated to (pH, temperature, salinity). Oxygenation may be advantageous, especially if hypoxia has been a problem under anesthesia. Assisted ventilation by rinsing gill surfaces with fresh, oxygenated water may be necessary until the fish completely recovers and manages on their own. Failure to attend to the animals’ needs during recovery from anesthesia will exacerbate and prolong the metabolic disturbances caused by anesthesia, and if seriously neglected, the fish may die. Mortality should be avoided.

### 2.14. Some Commonly Used Anesthetic Drugs in Fish

Care should be taken with selection of anesthetics as they do have significant side-effects and these should be researched to inform the appropriate choice of drug [40]. In principle, many of the drugs used for balanced anesthesia for other animals, such as ketamine [32], opioids, and NSAIDs [5], can be used for fish, but as they will demand manual handling to be administered, anesthetics that can be dissolved in and administered through the water are more commonly used.
**Active Substance****Characteristics and Use**Tricaine [35]MS222, Finquel^®^Anesthetic with analgesic effectsFast induction and recoveryAvailable as a powder to be dissolved in waterUsed for fish in fresh water; it makes an acidic pH, and buffering of water is therefore necessaryBenzocain [35]Benzoak^®^Anesthetic and sedativeLower safety margin at high temperaturesIsoeugenol [35]Aqui-S^®^Sedation for transport, handling, sea lice counting, or vaccinationMetomidate [41]Aquacalm^®^Short-acting hypnoticPoor analgesic effectLittle cortisol rise or cardiovascular effectProviding unconsciousnessSome commonly used anesthesia substances used for fish and their characteristics

## 3. Conclusions

Principles that apply for anesthesia and pain management in land-living mammals should be also adapted and adhered to for aquatic animals. These principles are important to support vital physiological functions. They are important for animal welfare reasons as well as for outcome of the procedures, and if neglected, there is a risk of complications and a fatal outcome.

## Figures and Tables

**Figure 1 biology-11-01355-f001:**
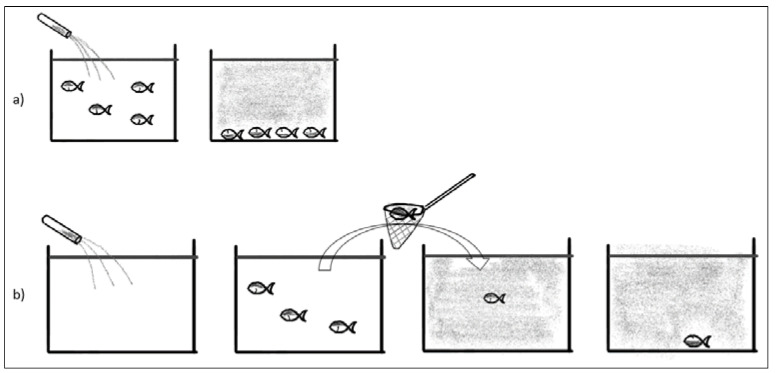
(**a**) Showing anesthesia solution directly into home tank. All fish will be anesthetized. (**b**) Individual fish are restrained and moved directly to the anesthesia bath.

**Figure 2 biology-11-01355-f002:**
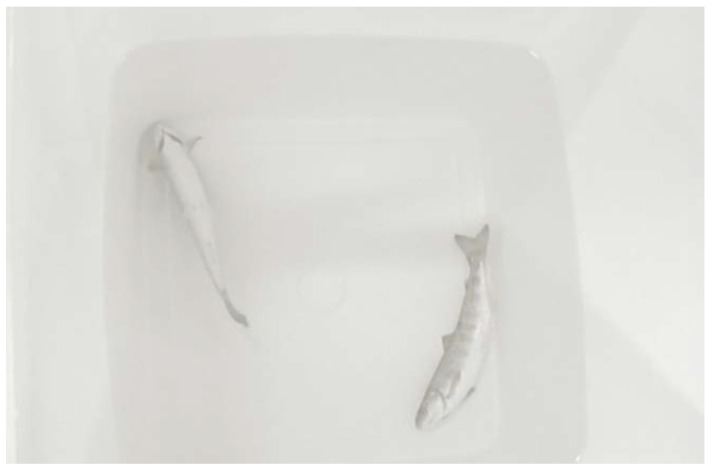
Anesthetized fish that have lost the righting up reflex and have sunk to the bottom of the tank, resting on their backs.

## Data Availability

Not applicable.

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
