# Peer review of "Good Anesthesia Practice for Fish and Other Aquatics"

_biology, 2022, doi:10.3390/biology11091355_

Round 1

Reviewer 1 Report

Dear Colleagues

many thanks for this nice guidelines which should wake up the stakeholders on the right of fish and aquatics to receive state of the art anesthesia and analgesia

My detailed comments in the document attached - mostly some rewording or consolidating the information present in the manuscript. Please take what you like and leave the rest

Kind regards

Author Response

Thank you for valuable feedback that I think improve the manuscript

My responses are included in the attached file

Reviewer 2 Report

The manuscript provides useful information for fish anesthesia, both for management purposes and for invasive procedures, such as surgery.

The text needs adjustments, which are mentioned in the sequence.

I recommend that authors check the punctuation and English of the entire text, as some flaws have been detected. For example, in line 76 the period at the end of the sentence is missing. In line 117 the word "that" appears repeated; In line 158, where the authors talk about temperature, only "2 C" appears, among other errors.

- Lines 93-94: “correct dose” - Add "/ concentration" since in many cases  the anesthetic is added to the water - “correct dose/ concentration”;

- Line 236: “below the safety level” - Please complement: "bellow the upper safety limit for animal products...";

- Lines 260-261: “Noise from”... - Please delate: "Noise from", starting the sentence with: "Pumps and...";

- P. 7: Add "Figure 2" before typing the title at the top of the page. Figure 2 should be cited in the text.

- Line 278: “...appealingly...” - Did you mean: "appearlingly"?

- Line 292: “..., hearth...” – Did you mean: "heart"?

- Line 300: “...where free gill surfaces for oxygen exchange is reduced” -  Did you mean: "are reduced"?

- Line 309: “...in anesthesia tank as fish is acclimatized....” - Did you mean: “...in anesthesia tank in which the fish are.."?

- Line 316: “...to be sure that the right actions timely taken.” - Did you mean: "...the right actions will be timely taken"?

- Line 323: “Avoid anesthesia in direct sunlight” - Note the wording of the other items, which describe causes of problems in anesthesia. I believe it would look better if this item were described as: "Anesthetic procedure performed in the presence of sunlight";

- Line 340: “ketamine’s” - Eliminate apostrophe, its use is not usual in scientific journals.

- In “Some commonly used anesthesia substances used for fish and their characteristics (32)”, in p. 8 – “Used in for fish in fresh water it makes an acidic Ph and buffering of water is therefore necessary”; Please correct: “Used for fish in fresh water it makes an acidic Ph and buffering of water is therefore necessary”;

- P. 9: “Slightly slower reduction” - What did you mean here? Please specify.

Author Response

(The authors gave the same response as above.)

Reviewer 3 Report

The current manuscript provides a practical guideline for anesthetic uses in fish with special emphasizing animal welfare and animal husbandry. The overall writing is good. The author explained the necessity of anesthetic application in appropriate detail. Nevertheless, there are some weak points that needed to be improved first. Please see my comments as follows.

Major comments
1.    The title “Good anesthesia practice” is too broad and can be misleading. In the current manuscript, the author mentioned anesthetics used in fish only and did not include human or other animals. So, the title may be changed to “Good anesthesia practice for fish welfare” or “Good anesthesia practice in fish aquaculture” or “Good anesthesia practice in fish management”, etc.
2.    The abstract is too short. The author please adds some major recommendations regarding the proper uses of anesthetic and appropriate protocols in fish research.
3.    The balance between the fish welfare and the results of the study should be mentioned since anesthetics itself can interfere with the overall experiment. For example, in the field of fish pharmacology and toxicology research, drug interactions between certain anesthetics and antibiotics (Rairat et al., 2021) or pollutants (Fabacher, 1982; Sijm et al., 1993) have been reported. In those cases, the effects of anesthetics should be aware.
References:
•    Fabacher, D.L., 1982. Hepatic microsomes from freshwater fish-II. Reduction of benzo(a)pyrene metabolism by the fish anesthetics quinaldine sulfate and tricaine. Comp Biochem Physiol C Comp Pharmacol. 73, 285-288.
•    Rairat, T., Chi, Y., Chang, S.K., Hsieh, C.Y., Chuchird, N., Chou, C.C., 2021. Differential effects of aquatic anaesthetics on the pharmacokinetics of antibiotics: Examples using florfenicol in Nile tilapia (Oreochromis niloticus). J Fish Dis. 44, 1579-1586.
•    Sijm, D.T.H.M., Bol, J., Seinen, W., Opperhuizen, A., 1993. Ethyl m-aminobenzoate methanesulfonate dependent and carrier dependent pharmacokinetics of extremely lipophilic compounds in rainbow trout. Arch Environ Contam Toxicol. 25, 102-109.
4.    Regarding the fish intended for human consumption (Line 233-240), the MRL and withdrawal time (WDT) for some important anesthetic (such as MS-222, benzocaine, eugenol, isoeugenol, etc.) should be provided. Note that different countries may have different regulations regarding the MRL and/or WDT. The author may select the regulation of one country as an example (such as Norway) or from several countries for comparison.
5.    The conclusion is lacking. The author please summarize the general recommendation of the anesthetic used in fish research.
6.    The reference number 32 “Fish anesthetics in Norway…” is a PowerPoint presentation. So, it is not appropriate for citation in the present context. Please use scientific papers or textbooks for such reference.

Other comments:
1.    Line 156-158: Please add relevant references, and 2 C should be replaced by 2°C.
2.    Line 178: The sentence “For some fish species fasting is a normal part of their life cycle” seems incomplete. The author may add the implication of fasting following this sentence. For example, “For some fish species fasting is a normal part of their life cycle. So, they may naturally well-tolerated starvation and a brief food deprivation might not severely harm the health of those fishes.”
3.    Line 266: “den” should be replaced by “then”.
4.    Figure 2: Please add the word “Figure 2” at the bottom of the picture.
5.    Line 292: “hearth” should be replaced by “heart”.
6.    Line 292-295: Please add relevant references.
7.    Line 340-341: NSAIDs are analgesics but not anesthetics. So, please remove it.
8.    Line 342: “solve” should be replaced by “dissolved”.
9.    Line 347: “this chapter” should be replaced by “this paper” or “this article”.
10.    Line 380: “og” should be replaced by “of”.
11.    Line 380: “anf” should be replaced by “and”.
12.    Line 382-384: The reference numbers 17-19 seem incomplete. There is no author(s), publishing company, country, etc.

Author Response

(The authors gave the same response as above.)

Reviewer 4 Report

The manuscript “Good anesthesia practice“ by Aurora Brønstad describes important measures around anesthesia in fish and other aquatic animals. This is an important topic as often steps beside the main application of the drug are not considered properly. Many scientists are no veterinarians and have never learned the implications of anesthesia in animals. This paper high lightens many of the points to consider before, during and after anesthesia. Therefore, I think it is of great benefit for the community.

I mainly have small points that need to be checked. As I’m not a native English speaker by myself some of the points raised here may be irrelevant. I apologize if this is the case.

Line 30: I would assume that the two verbs (protect and state) should be written in the 3rd person singular (protects, states).

Line 130-164: The order of the paragraph is inconsistent: It starts with talking about acclimatization, followed by behavior and feeding, before again starting to talk again about acclimatization and again feeding. That should be rearranged.

Line 158: the sign for degree is missing between “2  C”.

Line 223: Probably it should be “into the intestine or another internal organ”.

Lines 228-231: It would be of great help for readers to have more details and examples here on the mentioned substances for stimulation as it is important to have them available during the procedure.

Line 266: Probably it should be “Fish will then become inactive…”.

Figure above line 269: This should be numbered as Figure 2 and a refence in the text should be added.

Lines 278 (twice), 283 and in the table (twice): analgetic should be replaced by analgesic, as far as I know. This is the more common way of spelling when I look into the dictionaries.

Line 308: O2 should be changed in O2.

Line 316: the verb is missing in the last part of the sentence: “… that the right actions are timely taken.”

Line 321: The topic of humidification is sometimes mentioned in the text, but is nowhere explained why it is so important. I suggest few sentences about this topic in the text.

Table line with Tricaine: Ph should be replaced by pH.

Lines 380-384: The citations 16-19 lack bibliographical data (legal text reference or web source etc.)

Author Response

(The authors gave the same response as above.)

Round 2

Reviewer 3 Report

Biology-1817638 R1

The current manuscript has a significant improvement over the previous version. However, I have only some minor suggestions for the author.
1.    Line 235-241: Regarding the issue of maximum residue limits (MRL), although the author said that this is a general paper and will not include the information on MRL value, at least adding some “references” of legal MRL values (maybe from Norway or European Union’s regulation) into the manuscript would be a benefit to general readers as well. Note that it is not necessary to add specific MRL values for all available anesthetics. Just providing one or more relevant official references is enough.
2.    Line 369: “solve” should be replaced by “dissolved”.
3.    Page 9 (the table): Please provide the relevant references for the information presented in the table.
4.    Page 9 (the table): For isoeugenol, the description of “Sedation for example for transport” is better replaced by “Sedation for transport”.

Author Response

Comment Response
1.    Line 235-241: Regarding the issue of maximum residue limits (MRL), although the author said that this is a general paper and will not include the information on MRL value, at least adding some “references” of legal MRL values (maybe from Norway or European Union’s regulation) into the manuscript would be a benefit to general readers as well. Note that it is not necessary to add specific MRL values for all available anesthetics. Just providing one or more relevant official references is enough.

Examples and references are now included in the revised version of the manuscript

L 235-252

Line 369: “solve” should be replaced by “dissolved”.

Corrected in the revised version of the manuscript

L 380 
For isoeugenol, the description of “Sedation for example for transport” is better replaced by “Sedation for transport”.

Text has been rewritten in the revised version

Line 382-383